# Gold Leaf-Based Microfluidic Platform for Detection of Essential Oils Using Impedance Spectroscopy

**DOI:** 10.3390/bios12121169

**Published:** 2022-12-14

**Authors:** Ankita Sinha, Adrian K. Stavrakis, Mitar Simić, Sanja Kojić, Goran M. Stojanović

**Affiliations:** Faculty of Technical Sciences, University of Novi Sad, Trg Dositeja Obradovića 6, 21000 Novi Sad, Serbia

**Keywords:** peppermint, eucalyptus, essential oils, electrical resistance, microfluidic chip, oral dentistry

## Abstract

Drug delivery systems are engineered platforms for the controlled release of various therapeutic agents. This paper presents a conductive gold leaf-based microfluidic platform fabricated using xurography technique for its potential implication in controlled drug delivery operations. To demonstrate this, peppermint and eucalyptus essential oils (EOs) were selected as target fluids, which are best known for their medicinal properties in the field of dentistry. The work takes advantage of the high conductivity of the gold leaf, and thus, the response characteristics of the microfluidic chip are studied using electrochemical impedance spectroscopy (EIS) upon injecting EOs into its micro-channels. The effect of the exposure time of the chip to different concentrations (1% and 5%) of EOs was analyzed, and change in electrical resistance was measured at different time intervals of 0 h (the time of injection), 22 h, and 46 h. It was observed that our fabricated device demonstrated higher values of electrical resistance when exposed to EOs for longer times. Moreover, eucalyptus oil had stronger degradable effects on the chip, which resulted in higher electrical resistance than that of peppermint. 1% and 5% of Eucalyptus oil showed an electrical resistance of 1.79 kΩ and 1.45 kΩ at 10 kHz, while 1% and 5% of peppermint oil showed 1.26 kΩ and 1.07 kΩ of electrical resistance at 10 kHz respectively. The findings obtained in this paper are beneficial for designing suitable microfluidic devices to expand their applications for various biomedical purposes.

## 1. Introduction

Microfluidic devices have recently attracted immense research interest from the scientific community, specifically for lab-on-chip applications [1,2]. These devices possess numerous assets for their use in sensing applications, such as miniaturized size, cost-effectiveness, low sample consumption, well-controlled microenvironments, and ease of fabrication. Moreover, fast reaction time, high-throughput analysis, and portability are several other benefits which encourage their utilization over conventional sensing systems [2,3,4]. Interestingly, microfluidic lab-on-a-chip devices are currently showcasing a number of opportunities for controlled drug delivery applications where they can easily be automated to deliver therapeutic compounds, for example, drugs, to the target sites with precise control by integration, implantation, and localization [5,6]. The amount of fluid required in the miniaturized channels is very small, usually from nano to pico liters which makes these devices a favorable technique for such applications [7]. Therefore, facile preparation of microfluidic devices is of high significance in recent times, which in turn strongly depends on the choice of substrate material and the adopted methodology.

Early stages involved silicon and glass as the commonly applied substrates in preparing microfluidic devices [8]. However, newer low-cost, flexible substrates like polymer materials, paper, hydrogels and their various composites are now act as the interesting alternatives [9]. There are several reported techniques that have been developed over the years for fabricating microfluidic chips, such as lithography, laser polymer cutting, low-temperature co-fired ceramic technology (LTCC), injection molding and 3D printing (inkjet, screen, wax) [5,6,10]. For example, a polymer microchip-based microlens array was fabricated by injection molding for the detection of *S*. *typhimurium* in blood using immunoassays. However, high cost and the often requirement for a clean-room environment led to the search for more efficient approaches for microfluidic system fabrication. Unlike other methods, xurography is a low-cost method for the preparation of microfluidic devices without any specific requirement of complicated instrumentation or clean room facility [11,12,13]. Polymer and paper materials are the commonly utilized substrates in this method, reducing the cost of the overall fabrication procedure. Furthermore, easy handling and machine optimization operations related to adjusting force, velocity, and acceleration are other benefits which make xurography highly feasible and efficient in the preparation of microfluidic devices [14,15,16,17,18].

In the past few years, sensing performances of microfluidic devices comprising different substrate materials have been reported towards a varying range of targets by methods such as colorimetry, fluorescence, electrochemical, and surface-enhanced Raman spectroscopy [19]. With the advantages of precise fluid control within the channels and rapid response of these devices, the present work reports the construction of a gold (Au) leaf-based microfluidic device by xurography, where its electrical performance was evaluated using electrochemical impedance spectroscopy (EIS) under the influence of different essential oils (EOs). These oils are plant-based products which possess several anti-inflammatory, sedative, anxiolytic, antimicrobial properties, etc., and are extensively used as oral medicines [20,21,22,23,24,25]. As EOs are extensively applied in pharmaceutical industries, the present study encourages the potential utilization of microfluidic devices for controlled drug delivery applications in dentistry. EIS is considered one of the promising techniques that have been utilized in the study of drug administration into the human body by electrical characterization of target tissues [26,27]. Targeted drug delivery is a typical example of a dynamic, multidimensional process where EIS measurements can have an important role in real-time analysis of drug dispersion dynamics for improving the effectiveness of medical treatments. Moreover, electrical impedance analysis can be helpful for determining the performance of drug delivery devices under the influence of target drugs over a span of time by analyzing their electrical properties, which are useful for studying the feasibility of such devices. Taking inspiration from literature where EIS has been widely studied for such applications [26,27,28,29], the present study will utilize the technique for the electrical characterization of the fabricated microfluidic prototype, which is intended to be applied as a drug delivery system where EO act as a therapeutic drug. The present study demonstrates the extension of the work [29] where authors carry out extensive research towards studying the behavior of microfluidic chips toward drug delivery applications taking EOs as targets. However, the current work utilizes conductive gold leaf as electrode materials where the electrical responses of the microfluidic chip were studied up to a span of 46 h, while in the previous study, conductive silver-coated polyamide threads were used as electrode materials and resistance responses of the microfluidic chip were analyzed using different EOs (e.g., tea tree, clary sage, and cinnamon bark oil) over 60 min.

In this work, a straightforward EIS analysis was performed where exposure of drug (here EOs) onto the performance of delivery channels (within the proposed microfluidic prototype) connected to external electrodes was studied over a span of time. Au leaf was selected to be utilized as external electrodes because of its low resistivity (ρ ≈ 2.44 × 10^−8^ Ω·m), while peppermint and eucalyptus EOs were chosen as target drugs (sample fluids) due to their good electrical properties [30,31,32,33]. The performance of the microfluidic chip was determined by analyzing the change in its electrical resistance over a wide frequency range from 1 Hz to 100 kHz. Herein, the variation in the resistance of the device was studied under three parameters which were: (i) the effect of exposure time of the device to EOs up to a period of 46 h, (ii) the effect of different concentration (1% and 5%) of peppermint and eucalyptus oils, and lastly (iii) the effect of electrical properties of individual EOs. The work successfully presents a proof-of-concept microfluidic prototype where it aims at studying the feasibility of the device for drug delivery applications which was studied under the effect of EOs (targeted drugs) over a calculated time span.

## 2. Experimental

### 2.1. Materials and Apparatus

For building the microfluidic device, standard ISO A4 polyvinyl chloride (PVC) lamination foils of 80 µm and 125 µm thickness were used (MBL^®^ 80MIC and MBL^®^ 125MIC, Minoan Binding Laminating doo, Belgrade, Serbia). For the central chip layer, Au leaf (22 carats, purity 91%, >10 µm) was obtained from NRR, Schwabach, Germany. Peppermint and eucalyptus EOs were provided by Oshadi brand, AYUS, Bühl, Germany. In addition, deionized water (Grade 2 by ISO 3696, 1987) and ethanol (Laboratorija doo, Novi Sad, Serbia) were utilized to prepare different concentrations of essential oil solutions. A Cutter plotter (CE6000-60 PLUS, Graphtec America, Inc., Irvine, CA, USA) with a 45° cutting blade (CB09U) was used to cut the PVC foils. Bonding of different components was performed using hot lamination with an A4 card laminator (FG320, Minoan Binding Laminating doo, Belgrade, Serbia), and the designing of the microfluidic device was carried out using AutoCAD (Autodesk, 2021). The cross-sectional thickness of Au leaf was investigated by scanning electron microscopy (SEM) using TM3030, Hitachi (Chiyoda, Japan), at an accelerating voltage of 10 kV. Fourier transform infrared (FTIR) spectrum of Au leaf was performed in the spectral range 400–4000 cm^−1^ using ALPHA (Bruker, Bremen, Germany) in Attenuated Total Reflectance (ATR) mode. The Nanoindentation testing is performed with the Nano Indenter G200 (Agilent, Santa Clara, CA, USA) with the Berkovich tip, a load of 10 mN, peak hold time 1 s, unload up to 90%, and 100 measurement points per sample.

### 2.2. Fabrication of Microfluidic Device

The microfluidic device consists of basically a chip, and a three-step approach was followed to prepare it. Initially, the chip was designed, followed by cutting the micro-patterns using a cutter plotter, and lastly, by bonding different layers through hot lamination. The chip consists of four PVC foil layers with a thickness of 80 µm, two PVC layers with a thickness of 125 µm, one central layer of Au conductive strip, and two protective layers of PVC foil of thickness of 80 µm with a total chip dimension of 30 mm × 50 mm. The design of different layers of the fabricated microfluidic chip is shown in the first step in Figure 1. The diameter of the holes for the inlet and outlet was 1.25 mm. The dimension of the microfluidic channel in the middle of the structure was 11.53 mm × 2.5 mm. The dimension of the Au conductive strip was 20 mm × 5 mm, and it was used for measuring electrical parameters (real and imaginary parts of impedance). The final appearance of the prepared chip and the overall step toward their impedance analysis under the influence of EOs is shown in Figure 1. Moreover, different operating parameters of the cutter plotter and laminator are shown in Table 1.

### 2.3. Electrochemical Measurements and EO Sample Preparation

The fabricated Au leaf-based microfluidic chip was characterized electrochemically by EIS using Palmsense4 (PalmSens BV, Houten, the Netherlands) and PSTrace 5.8 software. The change in electrical resistance of the microfluidic device was studied by evaluating the real part of impedance (Z’) as a function of frequency in the range from 1 Hz to 100 kHz. To evaluate the electrical performance, the impedance of the dry chip was initially obtained, followed by measurements of samples with 1% and 5% of peppermint and eucalyptus EOs prepared in 10% ethanol solution containing distilled water.

## 3. Results and Discussion

### 3.1. Characterization of Au Leaf

Figure 2a represents the FTIR spectrum of Au leaf. The intense peak at around ~2900 cm^−1^ indicated the presence of C-H stretching of alkane groups, while peaks at 1750 cm^−1^ and 1450 cm^−1^ represented the C-C stretching of the aromatic ring and symmetric stretching of N-O group, respectively. Bands around 1250 cm^−1^ and 1030 cm^−1^ corresponded to C–N stretched aromatic amines and aliphatic amines, respectively. The spectrum clearly shows the organic origin of the Au sheet with added functionalities in the range from 3000 cm^−1^ to 500 cm^−1^ and matches well with previously published data [34]. Moreover, Figure 2b shows the SEM of the layer of Au leaf, revealing an observed thickness of 9 µm.

Figure 3 shows the nanoindentation measurements of the microfluidic chip where Figure 3a–d are nanoindentation load vs. displacement curves of electrode surface after 48 h exposure of liquid samples. The higher concentration of EO, regardless of its type, resulted in greater dispersion of load vs. displacement curves, which is the consequence of mild degradation of the Au layer and its adhesion to the PVC layer. Figure 3e shows curves of the Au layer before inserting any liquid sample, and Figure 3f shows the averaged curves of all measurements. After exposing the Au electrode to the liquid samples, the maximal indentation depth for the same load of 10 mN is significantly smaller than the depth reached on the electrode before exposure to the liquid.

### 3.2. EIS Studies

In the first segment of our study, the fabricated microfluidic prototype, i.e., the conductive gold strip, was exposed to peppermint and eucalyptus EOs up to a period of 46 h, and the change in electrical resistance was evaluated at different time intervals. The first measurement was performed at 0 h, which was immediately upon the injection of the EO solutions into the inlet of the microfluidic channel. After this measurement, the inlet and outlet of the microfluidic channel were sealed with parafilm to provide a hermetically sealed medium in the chip and to prevent fluid evaporation. The second measurement was performed 22 h after the first measurement, while the third measurement was performed 46 h after the first measurement. Thus, the electrical resistance (real part of complex impedance) of the gold conductive strip was measured under the influence of peppermint and eucalyptus EOs at 0 h, after 22 h, and after 46 h. For this, 12 µL of 1% and 5% essential oil samples were injected into the microchannels of the chip, and electrical resistance was measured at the two ends of the Au conductive electrode in the frequency range from 1 Hz to 100 kHz.

The obtained results are shown in Figure 4, which clearly demonstrates an increase in the electrical resistance of the microfluidic chip at 22 h and 46 h compared to 0 h and dry chip, suggesting a negative influence of EOs on the conductive strip at longer times which led to decrease of its conductivity. This trend was found to be similar for both tested concentrations (1% and 5%) of EOs. The reason for this change in resistance could be ascribed to the degradation of the gold strip at the part placed into the microfluidic channel (place of crossing) when exposed to essential oils for longer durations. The plots presented in Figure 4 demonstrate this behavior, as it is visible that the resistance of the chip is highest for the case of 46 h of exposure, followed by that for 22 h, and finally for 0 h. It can also be concluded from Figure 4 that Z’ value for the gold electrode is in the range from 0.8 Ω to 1.79 Ω, confirming the excellent conductivity of the gold-leaf strip, even after exposure to different EOs.

In the next step of our study, we investigated the influence of different concentrations of EO solution on the electrical resistance of the microfluidic chip. Figure 5 shows the change in electrical resistance as a function of frequency for 1% and 5% solutions of peppermint and eucalyptus EOs respectively. The graphs represent variation in resistance of microfluidic chip after 46 h exposure to different concentrations of EOs in the microfluidic channel.

These results suggest that under the influence of 1% EO solutions, the microfluidic chip overall demonstrated higher resistance than that of the 5% solution. This observation was found common in the case of both peppermint and eucalyptus EOs [35,36]. However, this study is a matter of further investigation where the resistance vs. concentration relationship can have different responses according to the composition and concentration of the solvents used for preparing the EO dispersion and also as per the chemical constituents of the EOs. In the present case, Table 2 confirms higher resistance of the microfluidic chip in the case of 1% essential oil solution than 5% at a frequency of 10 kHz.

In another development of our study, the influence of EOs on the electrical performance of the microfluidic chip was studied. For this purpose, a given volume (12 µL) of the peppermint and eucalyptus EOs in the same concentration were injected into the microchannels of the chip individually, and change in the resistance was measured after 46 h of exposure. The higher electrical resistance of eucalyptus oil was observed compared to peppermint oil for 1% and 5% solutions. It can be clearly seen from Figure 6 that eucalyptus oil has a higher impact on the electrical resistance of the chip than peppermint oil [37].

### 3.3. Calculation of Root Mean Square Deviation (RMSD) as a Tool to Differentiate Impact of Essential Oil Type and Concentration

Using the measured resistance (*Z’*) of gold strips shown in Figure 4, Figure 5 and Figure 6, it is possible to visually distinguish when different EOs types and concentrations were applied. However, as in all combinations of EO type concentration and exposure time, the resistance is increased when compared to the dry chip, and thus an easier and more convenient way of differentiation between the dry chip (baseline) and sample under test was established using specific essential oil type, and concentration and calculation of root mean square deviation (RMSD) was performed. With RMSD calculation, it is possible to get a scalar value capable of providing information on how big the difference is between the sample under test and the dry chip:(1)$$\mathrm{RMSD}=\sum_{i=1}^{N}{\left[\frac{{\left(Re\left(Baseline_{i}\right)-Re\left(SUT_{i}\right)\right)}^{2}}{{\left(Re\left(Baseline_{i}\right)\right)}^{2}}\right]}^{\frac{1}{2}}$$

A lower RMSD value means that the sample is more similar to the dry chip. Therefore, that can be interpreted as a weaker impact on the gold strip. Calculated RMSD values in the frequency range of 1 Hz-10 kHz are summarized in Table 3.

The obtained values in Table 3 are in compliance with Figure 4, Figure 5 and Figure 6. As it can be seen in Figure 4a, 1% of eucalyptus after 22 h is similar to the resistance after 0 h, but after 46 h, it is much higher than the resistance of the dry chip. With RMSD calculation, it is possible to confirm that assumption as RMSD after 0 h and 22 h are 18.12 and 23.07, respectively. However, after 46 h, it increased to 47.88. In the case of 5% of Eucalyptus (Figure 4b), resistance after 0 h and 22 h are very close to the dry schip but lower than for 1% of eucalyptus, which was confirmed with significantly lower RMSD values (3.61 < 18.12 and 6.64 < 23.07).

Therefore, with the proposed platform is possible to detect the impact level of various EO types and concentrations, as well as exposure time, which is of great importance in various biomedical applications, including drug delivery systems.

## 4. Conclusions and Future Prospects

This work aimed at fabricating a gold leaf-based microfluidic device using xurography for analyzing the effect of EOs on its electrical properties using electrochemical impedance spectroscopy. The performance of the fabricated device was evaluated by measuring the change in its electrical resistance by injecting different concentrations (1% and 5%) of eucalyptus and peppermint EOs into the microchannel. The study demonstrated an increase in the electrical resistance of the chip with increased exposure time to EOs, measured at 0 h, 22 h, and 46 h. Moreover, the effect of concentration and individual properties of EOs on the electrical performance of the chip were also analyzed. The findings obtained in this paper are helpful in analyzing the effect of EOs on the fabricated microfluidic systems over the course of time that could provide a new approach for effective drug delivery in the field of dentistry. The work represents a novel proof-of-concept prototype; however, further studies are required for the complete atomization of the fabricated device in terms of developing a portable readout unit and flow rate control of the fluid inside the channels. Further work is needed to explore the design and components, optimization of fabrication and detection methodology, and analysis of the reactivity of various drugs within the system. Apart from the drug delivery applications, integrated microfluidic devices have been explored for various biomedical sensing applications such as pathogen detection, and single cell and cell culture analysis using impedance spectroscopy [38,39,40,41,42] (Table 4). The present work takes the initiative of detecting essential oils, which are also gaining enormous attention in medical applications suggesting a huge opportunity for their utilization in the coming time.

Despite rapid development in the field of microtechnology, numerous possibilities and challenges remain to be executed and accepted for the practical applicability of microfluidic devices toward drug delivery [1]. Firstly, complete automation and the complexity of bringing multiple operations within a single microfluidic system is itself a challenge. Along with this, mass production of such devices and lack of suitable design standards are other obstacles in the implementation of such devices towards medical applications, which need to be addressed and coordinated between different fields of engineering, and physical and biological sciences for commercial applications. Moreover, it would be interesting to develop implantable microfluidic devices for long-term medical treatment using materials with biodegradable properties. However, in spite of all challenges, a continued expansion of microfluidic technologies is expected in the coming time, turning the proof-of-concept prototypes into commercial systems and bringing solutions to various real-world problems.

## Figures and Tables

**Figure 1 biosensors-12-01169-f001:**
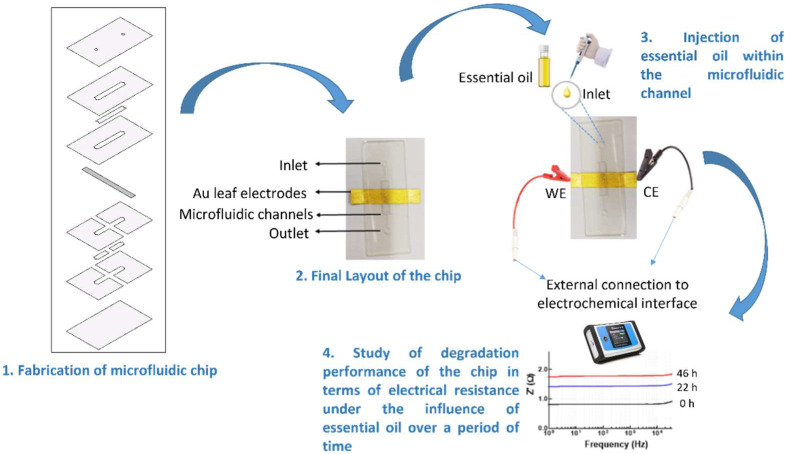
Exploded view of multi-layered Au leaf-based microfluidic chip design and layout of the study performed to analyze the electrical performance of the fabricated device under the influence of essential oil using impedance measurements.

**Figure 2 biosensors-12-01169-f002:**
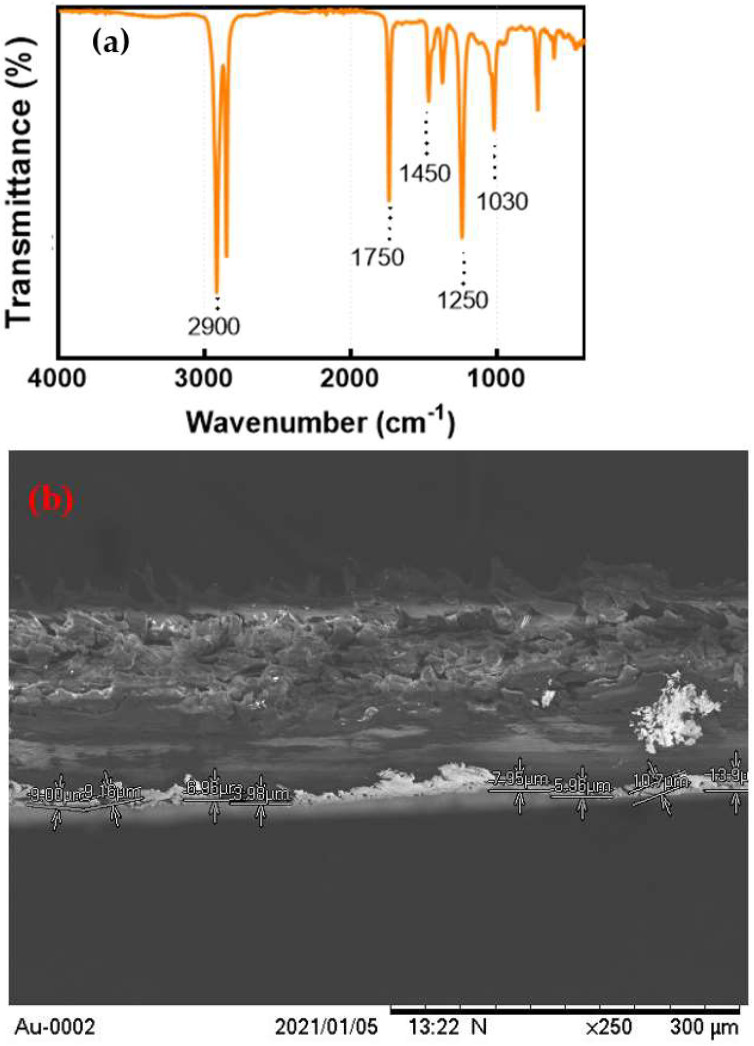
(**a**) FTIR spectrum and (**b**) SEM micrograph of the cross-section of Au leaf used for fabrication of microfluidic chip.

**Figure 3 biosensors-12-01169-f003:**
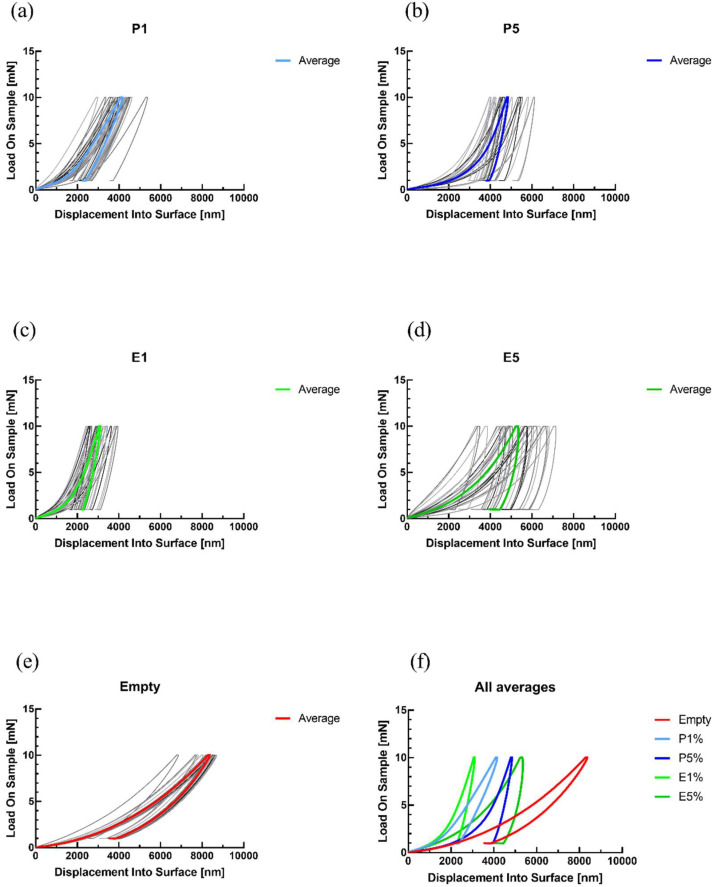
Nanoindentation results of microfluidic chip with (**a**–**d**) 1% and 5% of peppermint and eucalyptus essentials oils, (**e**) nanoindentation of dry chip before using any liquid sample, (**f**) average values of all samples.

**Figure 4 biosensors-12-01169-f004:**
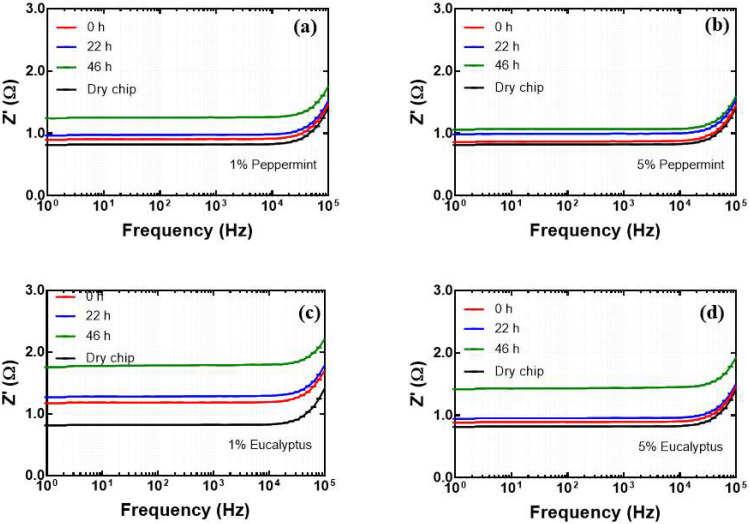
Electrical resistance of Au-based microfluidic chip as a function of frequency at different time intervals under the influence of (**a**) 1% peppermint, (**b**) 5% peppermint, (**c**) 1% eucalyptus, (**d**) 5% eucalyptus essential oils.

**Figure 5 biosensors-12-01169-f005:**
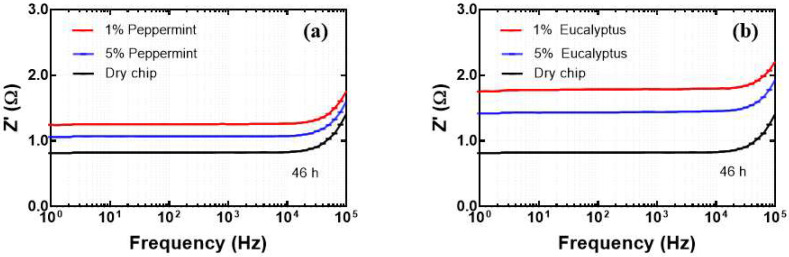
Electrical resistance of fabricated microfluidic chip for 1% and 5% concentrations of (**a**) peppermint and (**b**) eucalyptus EOs, after 46 h of exposure.

**Figure 6 biosensors-12-01169-f006:**
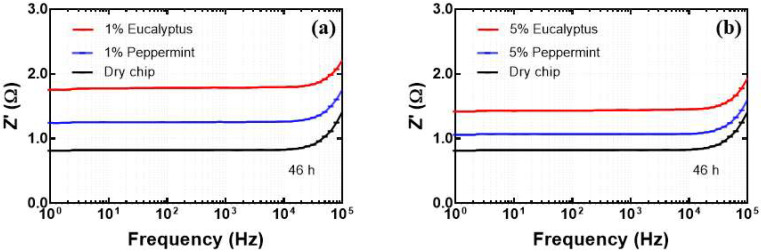
Change in resistance of fabricated microfluidic chip under the influence of (**a**) 1% and (**b**) 5% EOs solutions after 46 h of exposure.

**Table 1 biosensors-12-01169-t001:** Operating parameters of cutter plotter and hot laminator.

**Parameter**	**Value**
**Cutter plotter**	
Cutting speed	3 cm/s
Cutting force PVC 80 µm	19 (out of 38 steps from 0.2 to 4.41 N)
Cutting force PVC 125 µm	26 (out of 38 steps from 0.2 to 4.41 N)
**Hot lamination**	
Temperature	160 °C
Speed	3 cm/min

**Table 2 biosensors-12-01169-t002:** Effect of concentration of EOs on the electrical resistance of microfluidic chip at 10 kHz.

EOs in the Channel	Z’ [Ω] at 10 kHz 1%	Z’ [Ω] at 10 kHz 5%
Peppermint	1.26	1.07
Eucalyptus	1.79	1.45

**Table 3 biosensors-12-01169-t003:** Calculated RMSD values for EO types and concentrations at different exposure times.

EOs in the Channel	Time	RMSD 1%	RMSD 5%
Peppermint	0 h	4.29	2.39
Peppermint	After 22 h	7.73	8.57
Peppermint	After 46 h	21.62	12.33
Eucalyptus	0 h	18.12	3.61
Eucalyptus	After 22 h	23.07	6.64
Eucalyptus	After 46 h	47.88	30.65

**Table 4 biosensors-12-01169-t004:** Application of microfluidic systems in sensing using impedance measurements.

Substrate	Electrode Material	Method of Fabrication	Target	Reference
Polydimethoxy silane	Ti	Lithography	HEK29, HCT116 cell lines	[38]
Indium tin oxide-coated glass	Au	Photolithography	HeLa, NIH-3T3, and CHO-K cell culture	[39]
Glass	Au	Dielectrophoresis	*E. coli*	[40]
Glass	Au	Sputtering	Tomato Ringspot virus	[41]
Indium tin oxide-coated glass	Au	Photolithography	Human periph-eral blood mononuclear cell	[42]
Polyvinyl chloride	Ag	Xurography	EOs	[29]
Polyvinyl chloride	Au	Xurography	EOs	This work

## Data Availability

Data is available from the corresponding author upon request.

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
