# Peer review of "Gold Leaf-Based Microfluidic Platform for Detection of Essential Oils Using Impedance Spectroscopy"

_biosensors, 2022, doi:10.3390/bios12121169_

Round 1

Reviewer 1 Report

The manuscript by Sinha et al. reported an electrochemical microfluidic sensor for the detection of essential oils, which is expected to be used for biomedical engineering. Gold leaf with a high conductivity was used for chip fabrication, and the electrochemical properties have been extensively discussed. In my opinion, the question comes from the uncertainty of the research purpose. Why microfluidic chip was employed for this application? The authors should provide details of the potential application in drug delivery, and explain the significance of the results discussed in this manuscript. Besides, what is the significance of using impedance spectroscopy? How it can be relate to the application of essential oils in drug delivery? The authors should provide more detailed background information and its future prospects before further consideration of publication.  

Author Response

Reply to Reviewer 1 Comments:

General Comments: The manuscript by Sinha et al. reported an electrochemical microfluidic sensor for the detection of essential oils, which is expected to be used for biomedical engineering. Gold leaf with a high conductivity was used for chip fabrication, and the electrochemical properties have been extensively discussed. In my opinion, the question comes from the uncertainty of the research purpose.

Response: We thank the reviewer for the constructive comments to revise our manuscript. The authors have tried to implement the suggestions and to address all the queries raised by the reviewer to improve the quality of the manuscript. The manuscript has been now revised carefully according to the suggestions provided.

Comment 1: Why microfluidic chip was employed for this application? The authors should provide details of the potential application in drug delivery, and explain the significance of the results discussed in this manuscript.

Response: The possible potential and significance of the microfluidic chip towards the presented application in the field of drug delivery have been now discussed and included in the introduction section. Significance of the result is also discussed now in the introduction section.

“Microfluidic devices have recently attracted an immense research interest by the scientific community, specifically for lab-on-chip applications [1, 2]. These devices possess numerous assets for their use in sensing applications such as miniaturized size, cost effectiveness, low sample consumption, well-controlled microenvironments and ease of fabrication. Moreover, fast reaction time, high-throughput analysis and portability are several other benefits, which encourage their utilization over conventional sensing systems [2-4]. Interestingly, microfluidic lab-on-a-chip devices are currently showcasing a number of opportunities for controlled drug delivery applications where they can easily be automated to deliver therapeutic compounds, for example drugs, to the target sites with precise control by integration, implantation and localization [5, 6]. The amount of fluid required in the miniaturized channels is very small usually from nano to pico liters which makes these devices a favorable technique for such applications [7, 8]. Therefore, facile preparation of microfluidic devices is of high significance in recent time, which in turn strongly depends on the choice of substrate material and the adopted methodology.”

“Taking inspiration from literature where EIS has been widely studies for such applications [26-29], present study will utilize the technique for the electrical characterization of the fabricated microfluidic prototype which is intended to be applied as a drug delivery system where EO act as therapeutic drug.”

“In this work, a straightforward EIS analysis was performed where exposure of drug (here EOs) onto the performance of delivery channels (within the proposed microfluidic prototype) connected to external electrodes were studied over a span of time. Au leaf was selected to be utilized as external electrodes because of its low resistivity (r » 2.44·10−8 Ω·m), while peppermint and eucalyptus EOs were chosen as target drugs (sample fluids) due to their good electrical properties [30-33]. The performance of the microfluidic chip was determined by analyzing the change in its electrical resistance over a wide frequency range from 1 Hz to 100 kHz. Herein, the variation in the resistance of the device was studied under three parameters which were: (i) the effect of exposure time of the device to EOs up to a period of 46 h, (ii) the effect of different concentration (1% and 5%) of peppermint and eucalyptus oils, and lastly (iii) the effect of electrical properties of individual EOs. The work successfully presents a proof-of-concept microfluidic prototype where it aims at studying the feasibility of the device towards drug delivery applications which was studied under the effect of EOs (targeted drugs) over a calculated time span.”

Comment 2: Besides, what is the significance of using impedance spectroscopy? How it can be relate to the application of essential oils in drug delivery?

Response: The significance of using impedance spectroscopy for the present application has been included in the introduction section of the revised manuscript.

“EIS is considered as one of the promising techniques that has been utilized in the study of drug administration into human body by electrical characterization of target tissues [26, 27]. Targeted drug delivery is a typical example of a dynamic, multidimensional process where EIS measurements can have important role towards real-time analysis of drug dispersion dynamics for improving the effectiveness of medical treatments. Moreover, electrical impedance analysis can be helpful for determining the performance of drug delivery devices under the influence of target drugs over a span of time by analyzing its electrical properties which are useful to study the feasibility of such devices. Taking inspiration from literature where EIS has been widely studies for such applications [26-29], present study will utilize the technique for the electrical characterization of the fabricated microfluidic prototype which is intended to be applied as a drug delivery system where EO act as therapeutic drug.”

Comment 3: The authors should provide more detailed background information and its future prospects before further consideration of publication.  

Response: Significance of the present work has been discussed in the introduction section as suggested. The future prospects of the work is included now in the conclusion section.

“Despite rapid development in the field of microtechnology, numerous possibilities and challenges remains to be executed and accepted for practical applicability of microfluidic devices towards drug delivery [38]. Firstly, complete automation and the complexity of bringing multiple operations within a single microfluidic system is itself a challenge. Along with, mass production of such devices and lack of suitable design standards are other obstacles in the implementation of such devices towards medical applications which need to be addressed and coordinated between different fields of engineering, physical and biological sciences for commercial applications. Moreover, it would be interesting to develop implantable microfluidic devices for long-term medical treatment using materials of biodegradable properties. However, in spite of all challenges, a continued expansion of microfluidic technologies is expected in coming time turning the proof-of-concept prototypes into commercial systems and bringing solutions to various real-world problems.”

Reviewer 2 Report

Sinha A. et. al., have presents a conductive gold leaf-based microfluidic platform fabricated using xurography technique for controlled drug delivery. Moreover, the effect of the exposure time of chip to different concentrations of essential oils was analyzed and change in electrical resistance was measured at different time intervals. Compared to similar studies, it has some novelty, but the author should try to highlight them more. However, the author should also consider the following comments:

1.  Keywords should be changed completely.

2.     I could not find any results in the abstract. It is mainly about the technique which has been used. The author should discuss the results in the abstract as swell.

3.     Add more about similar published studies in the introduction.

4.     Line 80, close the parenthesis.

5.     Remove figure 1b since it does not has any scientific value.

6.     The Y-axis in 2a is wrong and should start from zero and not 0.3.

7.     The resolution for figure 2b is poor.

8.     Compare it with similar published literature in the discussion section.

9.     Rewrite the conclusion and show its significant

10.  . It is too short and empty of the main finding of this work. Add recommendations for future studies also.

Author Response

Reply to Reviewer 2 Comments:

General Comments: Sinha A. et. al., have presents a conductive gold leaf-based microfluidic platform fabricated using xurography technique for controlled drug delivery. Moreover, the effect of the exposure time of chip to different concentrations of essential oils was analyzed and change in electrical resistance was measured at different time intervals. Compared to similar studies, it has some novelty, but the author should try to highlight them more. However, the author should also consider the following comments.

Response: We thank the reviewer for the constructive comments to revise our manuscript. Utmost care has been taken to revise the manuscript.

Comment 1: Keywords should be changed completely.

Response: The keywords are now replaced in the revised manuscript.

Comment 2: I could not find any results in the abstract. It is mainly about the technique which has been used. The author should discuss the results in the abstract as well.

Response: Finding of the work are now briefly included in the abstract now.

“Moreover, eucalyptus oil had stronger degradable effects on the chip, which resulted in higher electrical resistance than that of peppermint. 1% and 5% of Eucalyptus oil showed an electrical resistance of 1.79 kΩ and 1.45 kΩ at 10 kHz while 1% and 5% of peppermint oil showed 1.26 kΩ and 1.07 kΩ of electrical resistance at 10 kHz respectively.”

Comment 3: Add more about similar published studies in the introduction.

Response: Introduction section has been revised. More literature and studied related to the background of the present work are now included.

Comment 4: Line 80, close the parenthesis.

Response: The correction has been made.

Comment 5: Remove figure 1b since it does not has any scientific value.

Response: The modification in Figure 1 has been made in the revised manuscript.

Comment 6: The Y-axis in 2a is wrong and should start from zero and not 0.3.

Response: The correction has been made in the revised manuscript related to Figure 2a.

Comment 7: The resolution for figure 2b is poor.

Response: Figure 2b is now replace with a new SEM micrograph in the revised manuscript.

Comment 8: Compare it with similar published literature in the discussion section.

Response: Comparison of the present work with other related published articles has been included now in the discussion section.

“This observation was found common in case of both peppermint and eucalyptus EOs which was also in well agreement with previous studies [35, 36] where resistance of the sensor decreased with increase in concentration of the solution under investigation, however, reports on analyzing the influence of a potential drug on such drug delivery devices are very rare.”

Comment 9: Rewrite the conclusion and show its significant

Response: Conclusion section has been revised.

“This work aimed at fabricating a gold leaf-based microfluidic device using xurography for analyzing the effect of EOs on its electrical properties using electrochemical impedance spectroscopy. The performance of the fabricated device was evaluated by measuring the change in its electrical resistance by injecting different concentrations (1% and 5%) of eucalyptus and peppermint EOs into the microchannel. The study demonstrated an increase in electrical resistance of the chip with increase in its exposure time to EOs which was measured at 0 h, 22 h and 46 h. Moreover, the effect of concentration and individual properties of EOs on the electrical performance of chip were also analyzed. The findings obtained in this paper are helpful to analyze the effect of EOs on the fabricated microfluidic systems over a course of time that could provide a new approach for effective drug delivery in the field of dentistry. The work represents a novel proof-of-concept prototype, however further studies are required for complete atomization of the fabricated device in terms of developing a portable readout unit and flow rate control of the fluid inside the channels. Further work is needed to explore the design and components, optimization of fabrication and detection methodology and analysis of reactivity of various drugs within the system.

Despite rapid development in the field of microtechnology, numerous possibilities and challenges remains to be executed and accepted for practical applicability of microfluidic devices towards drug delivery [38]. Firstly, complete automation and the complexity of bringing multiple operations within a single microfluidic system is itself a challenge. Along with, mass production of such devices and lack of suitable design standards are other obstacles in the implementation of such devices towards medical applications which need to be addressed and coordinated between different fields of engineering, physical and biological sciences for commercial applications. Moreover, it would be interesting to develop implantable microfluidic devices for long-term medical treatment using materials of biodegradable properties. However, in spite of all challenges, a continued expansion of microfluidic technologies is expected in coming time turning the proof-of-concept prototypes into commercial systems and bringing solutions to various real-world problems.”

Comment 10: It is too short and empty of the main finding of this work. Add recommendations for future studies also.

Response: Future recommendations and possibilities related to the studies have been now included in the conclusion section.

Reviewer 3 Report

I read this manuscript with interest. The following comments may help the authors to improve the manuscript before publishing.

1.      The author should mention the thickness of the final microfluidic chip.

2.       A list of materials and machines used in this work should be provided.

3.       A schematic for the set-up should be provided with details so that the readers can follow the experimental measurement (figure 1).

4.       What is the main reason to choose this fabrication method? There are other large scale fabrication methods such as polymer injection molding and hot embossing. See: https://pubs.acs.org/doi/abs/10.1021/acs.analchem.9b04863

The authors may need to discuss on the choice of fabrication method in the manuscript.

Author Response

Reply to Reviewer 3 Comments:

General Comments: I read this manuscript with interest. The following comments may help the authors to improve the manuscript before publishing.

Response: We appreciate the reviewer for the useful comments to improve the quality of our manuscript. All changes in the revised manuscript have been highlighted.

Comment 1: The author should mention the thickness of the final microfluidic chip.

Response: Total chip dimension of 30 mm ´ 50 mm. The details are already provided in section 2.2. Fabrication of microfluidic chip.

“The microfluidic device consists of basically a chip and a three-step approach was followed to prepare it. Initially, the chip was designed, followed by cutting the micro-patterns using a cutter plotter, and lastly by bonding different layers through hot lamination. The chip consists of four PVC foil layers with a thickness of 80 µm, two PVC layers with a thickness of 125 µm, one central layer of Au conductive strip and two protective layers of PVC foil of thickness 80 µm with a total chip dimension of 30 mm ´ 50 mm. Design of different layers of fabricated microfluidic chip is shown in Figure 1a. The diameter of the holes for inlet and outlet were 1.25 mm. Dimension of the microfluidic channel in the middle of the structure was 11.53 mm ´ 2.5 mm. Dimension of Au conductive strip was 20 mm ´ 5 mm and it was used for measuring electrical parameters (real and imaginary part of impedance).”

Comment 2: A list of materials and machines used in this work should be provided.

Response: List of materials and apparatus used in the present work has been already provided in detail in section 2.1 Materials and apparatus.

“For building the microfluidic device, standard ISO A4 polyvinyl chloride (PVC) lamination foils of 80 µm and 125 µm thickness were used (MBL® 80MIC and MBL® 125MIC, Minoan Binding Laminating doo, Belgrade, Serbia). For the central chip layer, Au leaf (22 carats, purity 91%, >10µm) was obtained from NRR, Germany. Peppermint and eucalyptus EOs were provided by Oshadi brand, AYUS, Germany. In addition, deionized water (Grade 2 by ISO 3696, 1987) and ethanol (Laboratorija doo, Novi Sad, Serbia) were utilized to prepare different concentrations of essential oil solutions. Cutter plotter (CE6000-60 PLUS, Graphtec America, Inc., USA) with a 45° cutting blade (CB09U) was used to cut the PVC foils. Bonding of different components was performed using hot lamination with an A4 card laminator (FG320, Minoan Binding Laminating doo, Serbia) and designing of microfluidic device was carried out using AutoCAD (Autodesk, 2021). The cross-sectional thickness of Au leaf was investigated by scanning electron microscopy (SEM) using TM3030, Hitachi, Japan at an accelerating voltage of 10 kV. Fourier transform infrared (FTIR) spectrum of Au leaf was performed in the spectrum range 400-4000 cm−1 using ALPHA (Bruker, Germany) in Attenuated Total Reflectance (ATR) mode. The Nanoindentation testing is performed with the Nano Indenter G200 (Agilent, USA) with the Berkovich tip, load of 10 mN, peak hold time 1 s, unload up to 90% and 100 measurement points per sample.”

Comment 3: A schematic for the set-up should be provided with details so that the readers can follow the experimental measurement (figure 1).

Response: A graphic representation of the present work has been now included as Figure 1.

Comment 4: What is the main reason to choose this fabrication method? There are other large scale fabrication methods such as polymer injection molding and hot embossing. See: https://pubs.acs.org/doi/abs/10.1021/acs.analchem.9b04863. The authors may need to discuss on the choice of fabrication method in the manuscript.

Response: The reason for the selection of xurography as a fabricaton method of the microfluidic chip in the present work is the low cost equipment and the components with user friendly approach that can easily be handled by even a nontechnical person. The choice of the method selection discussion is already included in the introduction section of the revised manuscript.

“There are several reported techniques which have been developed over the years for fabricating microfluidic chips such as lithography, laser polymer cutting, low temperature co-fired ceramic technology (LTCC), and 3D printing (inkjet, screen, wax) [5, 6]. However, high cost and the requirement for a clean-room environment led to the search for more efficient approaches. Unlike other methods, xurography is a low-cost method for preparation of microfluidic devices without any specific requirement of complicated instrumentation or clean room facility [11-13]. Polymer and paper materials are the commonly utilized substrates in this method reducing the cost of the overall fabrication procedure. Furthermore, easy handling and machine optimization operations related to adjusting force, velocity and acceleration are other benefits, which make xurography highly feasible and efficient towards preparation of microfluidic devices [14-18].”

Round 2

Reviewer 2 Report

The authors should provide a table at the end of the discussion section and compare their results with similar related published reports in order to show the significance of their study.

I would like to ask the authors to explain the novelty and enhancement they have in the current manuscript in comparison to their published article "Silver Thread-Based Microfluidic Platform for Detection of Essential Oils Using Impedance Spectroscopy" , Appl. Sci. 202212(7), 3596; https://doi.org/10.3390/app12073596.

Author Response

Reply to reviewer 2 comments:

  1. The authors should provide a table at the end of the discussion section and compare their results with similar related published reports in order to show the significance of their study.

Response: Table 4 has been now included in the revised manuscript as suggested.

Apart from the drug delivery applications, microfluidic integrated devices have been explored for various biomedical sensing applications such as pathogen detection, single cell and cell culture analysis using impedance spectroscopy [38-42] (Table 4). The present work takes the initiative of detecting essential oils which are also gaining enormous attention in medical applications suggesting a huge opportunities for their utilization in coming time.    

  1. I would like to ask the authors to explain the novelty and enhancement they have in the current manuscript in comparison to their published article "Silver Thread-Based Microfluidic Platform for Detection of Essential Oils Using Impedance Spectroscopy" ,  Sci.2022, 12(7), 3596; https://doi.org/10.3390/app12073596.

Response: The authors have now included the novelty statement in the revised manuscript in comparison to the said reference (now included as reference number 29).

The present study demonstrates the extension of the work [29] where authors carry out extensive research towards studying the behavior of microfluidic chip towards drug delivery applications taking EOs as targets. However, the current work utilizes conductive gold leaf as electrode materials where the electrical responses of the microfluidic chip was studied upto a span of 46 h while in the previous study, conductive silver-coated polyamide threads were used as electrode materials and resistance responses of the microfluidic chip were analyzed using different EOs (e.g. tea tree, clary sage, and cinnamon bark oil) over 60 min.   

Reviewer 3 Report

The authors did not discuss the reference in my comment.

Author Response

Reply to reviewer 3 comment:

  1. The authors did not discuss the reference in my comment.

Response: The reference has been now included as reference 10.
